# DA-DPO: Cost-efficient Difficulty-aware Preference Optimization for Reducing MLLM Hallucinations

**Longtian Qiu**  *qiult@shanghaitech.edu.cn*
*ShanghaiTech University, Shanghai, China*

**Shan Ning**  *ningshan2022@shanghaitech.edu.cn*
*ShanghaiTech University, Shanghai, China*
*Lingang Laboratory, Shanghai, China*

**Chuyu Zhang**  *zhangchy2@shanghaitech.edu.cn*
*ShanghaiTech University, Shanghai, China*

**Jiaxuan Sun**  *sunjx2022@shanghaitech.edu.cn*
*ShanghaiTech University, Shanghai, China*

**Xuming He**  *hexm@shanghaitech.edu.cn*
*ShanghaiTech University, Shanghai, China*
*Shanghai Engineering Research Center of Intelligent Vision and Imaging*

**Reviewed on OpenReview:** *https://openreview.net/forum?id=M52CgPcgGx*

## Abstract

Direct Preference Optimization (DPO) has shown strong potential for mitigating hallucinations in Multimodal Large Language Models (MLLMs). However, existing multimodal DPO approaches often suffer from overfitting due to the difficulty imbalance in preference data. Our analysis shows that MLLMs tend to overemphasize easily distinguishable preference pairs, which hinders fine-grained hallucination suppression and degrades overall performance. To address this issue, we propose Difficulty-Aware Direct Preference Optimization (DA-DPO), a cost-effective framework designed to balance the learning process. DA-DPO consists of two main components: (1)*Difficulty Estimation* leverages pre-trained vision–language models with complementary generative and contrastive objectives, whose outputs are integrated via a distribution-aware voting strategy to produce robust difficulty scores without additional training; and (2) *Difficulty-Aware Training* reweights preference pairs based on their estimated difficulty, down-weighting easy samples while emphasizing harder ones to alleviate overfitting. This framework enables more effective preference optimization by prioritizing challenging examples, without requiring new data or extra fine-tuning stages. Extensive experiments demonstrate that DA-DPO consistently improves multimodal preference optimization, yielding stronger robustness to hallucinations and better generalization across standard benchmarks, while remaining computationally efficient. The project page is available at `https://artanic30.github.io/project_pages/DA-DPO`.

## 1 Introduction

Recent advancements in Multimodal Large Language Models (MLLMs) (Liu et al., 2023a; OpenAI, 2023; Li et al., 2024) have significantly improved vision-language tasks, such as image captioning (Lin et al., 2014), visual question answering (Agrawal et al., 2015; Mathew et al., 2021; Marino et al., 2019). By combining powerful large language models with state-of-the-art vision models, MLLMs have enhanced multimodal understanding and reasoning. However, a persistent challenge for MLLMs is their tendency to produce

responses that are not reliably grounded in visual inputs, often resulting in "hallucinations" where descriptions include non-existent or inaccurate visual details. This limitation affects the reliability of MLLMs, posing a significant barrier for applications that require high factual accuracy.

Recent efforts have turned to Direct Preference Optimization (DPO)(Rafailov et al., 2024) as a promising approach to mitigate hallucinations in MLLMs. DPO encourages models to align their outputs with preference data that favor faithful responses and reduce hallucinations. Crucially, the effectiveness of DPO hinges on the quality of pairwise preference data. To address this, early approaches (Sun et al., 2023b; Yu et al., 2024a) rely on manual annotation, but such data collection is both labor-intensive and difficult to scale. More recently, several works (Pi et al., 2024; Li et al., 2023d; Zhou et al., 2024c; Yang et al., 2025) have proposed automated strategies for constructing multimodal preference data. These methods exploit trained models to produce pairwise preference data at scale, significantly increasing data coverage across diverse scenarios and thereby improving the model's ability to reduce hallucinations.

Despite their effectiveness in reducing hallucinations, vanilla DPO methods trained on existing pairwise preference data often lead to noticeable degradation in general multimodal capabilities, as shown in Figure 1a. We attribute this limitation to an imbalance between easy and hard samples in the training data, as illustrated in Figure 1b. Easy samples typically involve clearly distinguishable faithful and hallucinated responses, whereas hard samples require more nuanced reasoning to differentiate. This imbalance leads to a training bias where models overfit to easy cases while failing to learn from more challenging examples. We provide a detailed empirical analysis of this phenomenon in Section 3, showing that while models quickly adapt to easy samples, they struggle to generalize to hard ones, ultimately limiting the effectiveness of preference-based alignment.

To address this limitation, we propose a difficulty-aware training framework that dynamically balances the contribution of easy and hard samples during preference optimization. A key challenge in implementing this strategy is the *lack of explicit supervision for estimating sample difficulty.* We tackle this by introducing a lightweight, training-free strategy: by aggregating signals from multiple pre-trained vision–language models (VLMs) trained under diverse paradigms, we obtain robust difficulty scores that estimate the difficulty of pairwise preference data without explicit training a specific model. These difficulty scores are then used to reweight preference data, enabling effective difficulty-aware training that emphasizes harder samples while preventing overfitting to easier ones.

Specifically, we propose **D**ifficulty **A**ware **D**irect **P**reference **O**ptimization (DA-DPO), a framework that consists of two steps: *difficulty estimation* and *difficulty-aware training.* The first step assesses the difficulty of each pairwise preference sample using multiple VLMs. In particular, we leverage both contrastive VLMs (e.g., CLIP (Radford et al., 2021)) and generative VLMs (e.g., LLaVA (Liu et al., 2023b)) to estimate difficulty from complementary perspectives. Their outputs are aggregated through a distribution-aware voting strategy, in which the weight of each VLM is adaptively derived from its observed classification reliability over the training data. Building on these scores, the second step performs difficulty-aware training by dynamically adjusting the optimization strength of each sample in DPO. Specifically, the difficulty scores adjust the degree of divergence permitted between the learned policy and the initial policy. This mechanism strengthens learning from challenging samples while limiting unnecessary drift on trivial ones.

We conduct experiments on three popular MLLMs with different scales and abilities. To provide a comprehensive comparison, we report the performance comparison and analysis on two sets of benchmarks, hallucination benchmarks (Wang et al., 2023a; Rohrbach et al., 2018; Sun et al., 2023b; Li et al., 2023e) and general MLLM benchmarks (Hudson & Manning, 2019; Liu et al., 2023b; Fu et al., 2023; Li et al., 2023a), which demonstrate the effectiveness of our approach.

Our main contributions are summarized as follows:

- We conduct analysis on the multimodal preference optimization training and empirically demonstrate the existence of an overfitting issue, which can lead to suboptimal performance.

- We propose a cost-effective framework that leverages vision-language models (VLMs) to estimate the sample difficulty without additional training and utilize the estimation to improve preference modeling via difficulty-aware training.

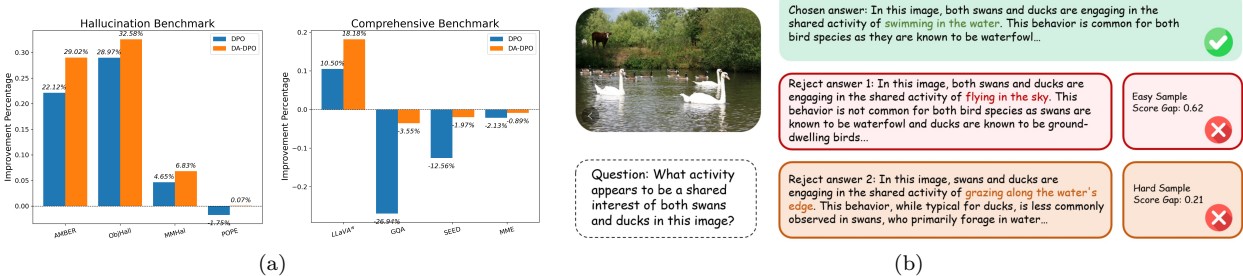

Figure 1: **(1a) Performance Comparison of DPO and DA-DPO.** We provide the performance improvements of DPO and DA-DPO compared to the LLaVA v1.5 7B without preference optimization. The *Hallucination* indicates the performance on 4 hallucination benchmarks, and *Comprehensive* indicates the performance of 4 comprehensive MLLM benchmarks. The details are described in the experiments section. **(1b) Easy and hard pairwise samples:** "Easy Samples" have a large score gap due to clear differences between preferred and dispreferred responses, while "Hard Samples" show minor differences, making them more valuable for learning.

- We evaluate our method on hallucination and comprehensive benchmarks, and experimental results show that it significantly enhances the performance of various MLLMs in a cost-efficient manner.

## 2 Preliminaries

In this section, we provide a brief overview of the Reinforcement Learning from Human Feedback (RLHF) to Direct Preference Optimization (DPO) pipelines.

**RLHF** Reinforcement Learning from Human Feedback (RLHF) is a widely used framework for aligning LLMs with human values and intentions. The standard approach (Bai et al., 2022a; Ouyang et al., 2022) first trains a reward model and then optimizes a KL-regularized reward objective to balance preference alignment with output diversity. The optimization can be written as:

$$\max_{\pi_\theta} \mathbb{E}_{x\sim D,\, y\sim \pi_\theta(x)}[r_\phi(x,y)] - \beta\, \mathrm{KL}[\pi_\theta(y|x)\,\|\,\pi_{\mathrm{ref}}(y|x)], \tag{1}$$

where $\pi_{\mathrm{ref}}$ is a reference policy (typically the SFT model) and $\beta$ controls the trade-off between reward maximization and staying close to $\pi_{\mathrm{ref}}$. The objective is usually optimized with PPO (Ouyang et al., 2022).

**Pair-wise Preference Optimization** Despite the success of the above RLHF, PPO is challenging to optimize. To enhance the efficiency of PPO, DPO (Rafailov et al., 2024) reparameterizes the reward function with the optimal policy:

$$r(x,y) = \beta \log\left(\frac{\pi_\theta(y\mid x)}{\pi_{\mathrm{ref}}(y\mid x)}\right) + \beta \log Z(x), \tag{2}$$

where $Z(x)$ denotes the partition function ensuring proper normalization. The hyperparameter $\beta$, analogous to the KL weight in Eq. (1), controls the trade-off: a larger value encourages $\pi_\theta$ to remain closer to the reference policy, preserving generalization and robustness, while a smaller value places greater emphasis on preference alignment but risks overfitting.

Building on this reward formulation, we can directly integrate it into the Bradley-Terry model, which treats pairwise preferences probabilistically. By doing so, we can optimize the preference objective without learning a separate reward model. The optimization objective is described as:

$$\min_{\pi_\theta} -\mathbb{E}_{x,y_c,y_r}[\log \sigma(r(x,y_c) - r(x,y_r))], \tag{3}$$

where $r(x,y)$ can be any reward function parametrized by $\pi_\theta$, such as those defined in Eq. ( 2) and $y_c$ and $y_r$ denote the chosen and rejected responses in pairwise preference data, respectively.

# 3 Multimodal Preference Optimization Analysis

In this section, we present a systematic investigation of the prevalent overfitting challenge in multimodal preference optimization. Through empirical analysis, we demonstrate that models exhibit a tendency to overfit to simpler training samples, while progressively reducing their effective learning from harder instances. This phenomenon is particularly pronounced in pairwise training paradigms like DPO (Rafailov et al., 2024). This overfitting behavior ultimately compromises model performance when applied to diverse real-world scenarios. We substantiate these findings with quantitative evidence drawn from training dynamics and reward trend analyses.

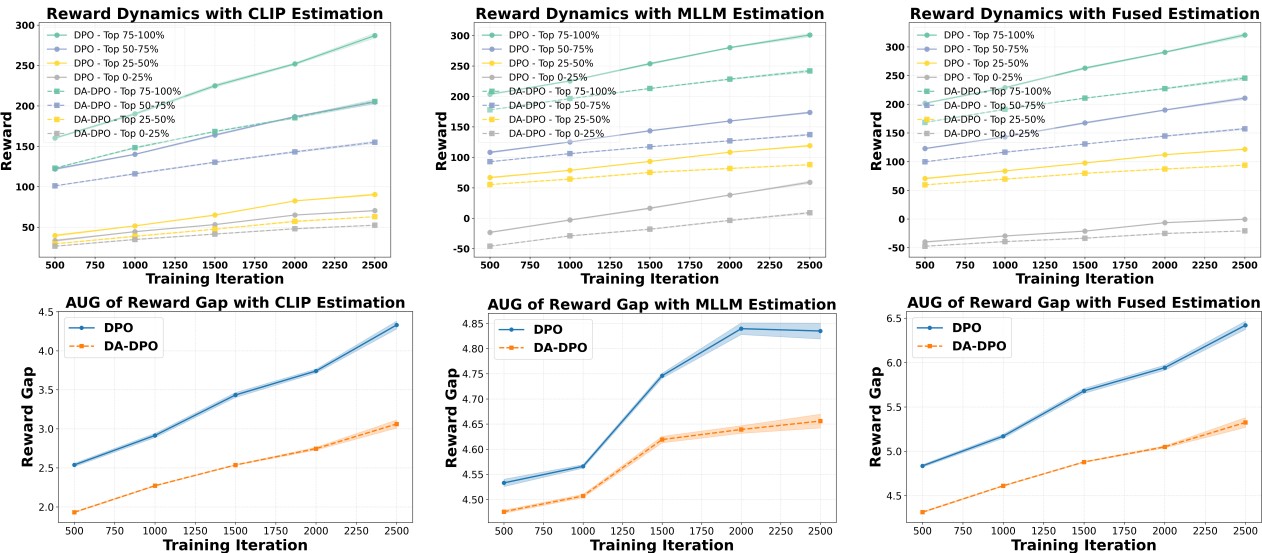

Figure 2: **Reward Dynamics and Area-Under-Gap (AUG) Between the Easiest and Hardest Samples.** We present the reward trajectories on a held-out validation set of LLaVA-v1.5-7B for both DPO and DA-DPO trained on the BPO dataset. The first row depicts how the rewards of data buckets with different estimated difficulty levels evolve over training iterations. The difficulty is estimated using three distinct proxies, as detailed in Section 4.1. Here, the *Easiest samples* (Top 75–100% in the legend) correspond to samples with the largest gap between chosen and rejected responses. The second row reports the *Area Under Gap* (AUG), quantifying the cumulative reward gap between the easiest and hardest samples across training, serving as a compact indicator of reward disparity dynamics. Shaded regions in the plots indicate the standard deviation across three independent seeds, though the effect is subtle due to the low variance induced by training randomness.

**Analysis Setting** To systematically analyze overfitting in preference optimization, we construct a controlled evaluation setup. We split the dataset into a training set and a held-out validation set with a ratio of 90% to 10%. Both DPO and DA-DPO models are trained on the 90% training portion, while their reward performance is periodically evaluated on the validation set across training iterations. Since there is no oracle annotation for sample difficulty, we estimate the difficulty of each validation sample using three different proxy metrics, as introduced in Section 4.1. Based on the difficulty ranking derived from each proxy, we partition the validation samples into four equally sized buckets, ranging from the easiest to the hardest. This setup allows us to examine how the model's reward evolves for samples of varying difficulty, thereby providing insights into overfitting behavior during preference optimization. To assess the statistical reliability of the results, we repeat the experiments with three different random seeds and report the corresponding standard deviations.

**Reward Dynamics Analysis** We analyze the reward dynamics from two complementary perspectives. The first perspective examines how the rewards of samples with different difficulty evolve throughout the training process. Based on three different proxy metrics (as detailed in Section 4.1), the validation samples

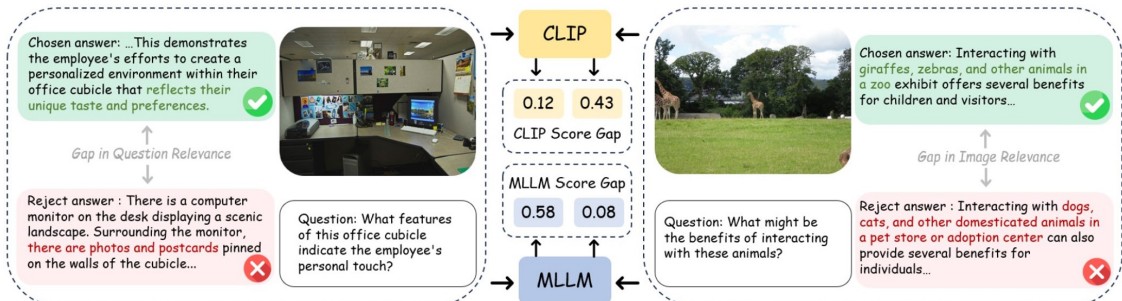

Figure 3: **Illustration of the two contrastive and generative VLMs.** As shown on the right side, CLIP captures the misalignment between the image and the answer such that the CLIP score gap is large when chosen and rejected answers differ in image relevance. However, as shown on the left side, the MLLM is better at capturing the logical connection between the question and answers, such that the MLLM score gap is large when chosen and rejected answers differ in question relevance.

are ranked by difficulty and partitioned into multiple buckets, from the easiest to the hardest. As shown in the first row of Figure 2, we observe that for both DPO and our proposed DA-DPO, the rewards of harder buckets consistently remain lower than those of easier buckets. This trend indicates a limited capacity to fit hard samples, which typically require fine-grained understanding capabilities. Moreover, *we observe that in DA-DPO, the reward of the easier buckets increases more slowly compared to that in naive DPO*. This slower growth is a result of DA-DPO's adaptive weighting mechanism, which adjusts the importance of each training instance based on its estimated difficulty. By doing so, DA-DPO effectively reduces over-optimization on easier samples, thereby alleviating the overfitting tendency frequently observed in standard DPO training.

To further quantify this phenomenon, we introduce a second perspective: the *Area Under Gap* (AUG) between the easiest and hardest samples. The AUG is computed by first taking the difference between the reward of the easiest bucket and that of the hardest bucket at each evaluation point across training iterations, and then integrating this gap over the entire training trajectory. This provides a cumulative measure of the reward disparity between easy and hard samples throughout training. A larger AUG indicates a stronger optimization bias toward easier samples. As shown in the second row of Figure 2, the AUG in DA-DPO remains consistently smaller than that in naive DPO, demonstrating that DA-DPO achieves a more balanced optimization across samples of varying difficulty. These analyses collectively highlight the presence of overfitting in multimodal preference optimization and the effacacy of DA-DPO in mitigating it.

## 4    Methods

In this section, we introduce the DA-DPO framework, which addresses the overfitting issue in standard multimodal DPO training in a cost-efficient manner. In Section 4.1, we first discuss the estimation of preference data, where CLIP and MLLMs to evaluate the difficulty of preference data. In Section 4.2, we explain how the data difficulty estimation from pretrained VLMs informs and guides the difficulty-aware DPO training.

### 4.1    Data Difficulty Estimation

The key challenge in evaluating the difficulty of preference data is the lack of explicit supervision. To address this, we propose a lightweight, training-free strategy that leverages pre-trained contrastive and generative VLMs to estimate sample difficulty from complementary perspectives, as illustrated in Figure 3. To combine the estimates from multiple models, we adopt a distribution-aware voting strategy, where each model's contribution is proportional to its preference classification accuracy on the training set. This results in a robust difficulty score for each sample without requiring additional model training. The details are described in the following sections.

**Contrastive Estimation**    CLIP is trained on web-scale image captions via contrastive training objectives and is proven to contain generalized knowledge regarding image and text relevance. We utilize CLIP to evaluate the difficulty of pairwise DPO data. Specifically, we first compute the CLIP text embeddings for the chosen response $y_c$ and rejected response $y_r$ (denoted as $f_c$ and $f_r$, respectively), and the CLIP image embedding for the image in DPO data $m$ (denoted as $v_m$). The CLIP scores $c_c$ and $c_r$ represent the image-text relevance of both responses is computed as follows:

$$c_c = \text{CosSim}(f_c, v_m), \quad c_r = \text{CosSim}(f_r, v_m). \tag{4}$$

We then introduce CLIP score gap $c_g$, which reflects the difficulty of a DPO sample; a larger gap indicates that the chosen and rejected responses are easily distinguishable in terms of image-text relevance. Formally, the CLIP score gap $c_g$ is defined as follows:

$$c_g = c_c - c_r. \tag{5}$$

To integrate this metric with other VLM-based estimates, we normalize $c_g$ to a common scale via dataset-level Gaussian projection. Given a dataset of $N$ samples, the normalized score is computed as follows:

$$\hat{c}_g = \Phi(\frac{c_g - \mu_{c_g}}{\sigma_{c_g}}), \tag{6}$$

where $\mu_{c_g}$ and $\sigma_{c_g}$ are the mean and variance of all CLIP score gaps $c_g$ in the DPO dataset. The normalized CLIP score gap $\hat{c}_g \in [0, 1]$, and $\Phi(\cdot)$ denotes the cumulative distribution function of the standard Gaussian distribution.

**Generative Estimation**    Recent Multimodal large language models (MLLMs) are built on large language models (LLMs) and learn to interact with the visual world by connecting a visual encoder to an LLM with language modeling objectives. For simplification, we denote this as a generative VLM. Although such models are prone to hallucination, they provide an informative evaluation of DPO data difficulty from another perspective complementary to CLIP. We extract the MLLM score for the chosen responses $y_c$ and rejected responses $y_r$, denoted as $m_c \in \mathbb{R}$ and $m_r \in \mathbb{R}$, which is defined as follows:

$$m_c = \sum_{t=1}^{T} \log\left(\text{P}(y_c^t | m, y_c^1, \dots, y_c^{t-1}; \pi_{\text{ref}})\right), \tag{7}$$

$$m_r = \sum_{t=1}^{T} \log\left(\text{P}(y_r^t | m, y_r^1, \dots, y_r^{t-1}; \pi_{\text{ref}})\right), \tag{8}$$

where $y_c^t$ and $y_r^t$ are the $t^{\text{th}}$ tokens in the chosen and rejected responses, and $T$ is the sequence length. To evaluate the difficulty of pairwise DPO data, the MLLM score gap $m_g$ is computed as follows:

$$m_g = m_c - m_r. \tag{9}$$

For a similar reason as the normalization of CLIP score, we utilize a Gaussian normalization to acquire a normalized MLLM score gap $\hat{m}_g \in [0, 1]$. The process is defined as follows:

$$\hat{m}_g = \Phi(\frac{m_g - \mu_{m_g}}{\sigma_{m_g}}), \tag{10}$$

where $\mu_{m_g}, \sigma_{m_g}$ is the mean and variance of all the MLLM score gaps $m_g$ in the DPO dataset.

**Distribution-aware Voting Fusion**    To this end, we evaluate the pairwise DPO data from two perspectives, resulting in two difficulty scores. As shown in Figure 4, the two VLMs perform differently in preference classification: CLIP excels on caption-related preference data, while MLLM performs better on VQA-related

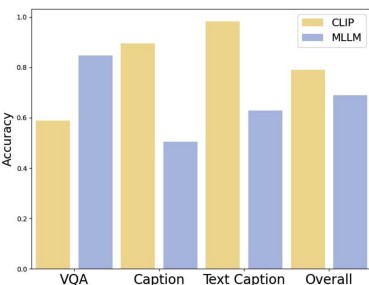

Figure 4: **Preference classification comparison.** Preference classification evaluates whether the pre-trained VLMs output a higher reward score for the chosen answer compared with the rejected answer. We report the classification accuracy on three subcategories and the overall performance on the BPO training dataset.

data. We propose a data-driven voting strategy to adaptively combine the difficulty scores based on the preference classification results. Specifically, we use the classification accuracies of CLIP and MLLM, denoted as $cls_c$ and $cls_m$, to determine the weight of *beta* for each DPO sample, as described below:

$$\hat{\beta} = ( \frac{cls_c}{cls_c + cls_m} \hat{c}_g + \frac{cls_m}{cls_c + cls_m} \hat{m}_g ) + 1, \tag{11}$$

where we add a constant term of 1 to improve the stability of the training process. Without this term, when both $\hat{c}_g$ and $\hat{m}_g$ are close to zero, the resulting $\hat{\beta}$ would also approach zero, potentially leading to training collapse.

### 4.2 Difficulty-aware Training

After estimating the difficulty of the preference pairs, we obtain a robust score that reflects the difficulty of the pairwise DPO data. Building on previous work (Wu et al., 2024), we perform difficulty-aware DPO training by adaptively calibrating the $\beta$ in Eq. (1). This approach allows us to adjust the weight of each training sample, reducing overfitting on easy samples compared to standard DPO training. Formally, the proposed difficulty-aware DPO objective is defined as:

$$\mathcal{L}_{\text{DA-DPO}} = -\mathbb{E}_{(x,y_c,y_r,\hat{\beta}) \sim D} \left[ \log \sigma \left( \hat{\beta} \, r(x, y_c) - \hat{\beta} \, r(x, y_r) \right) \right], \tag{12}$$

where $\hat{\beta}$ is the difficulty-aware scaling factor, $r(x, y)$ denotes the reward function as defined in Eq. (2), which itself incorporates the temperature coefficient $\beta$, $y_c$ and $y_r$ are the chosen and rejected responses, respectively, and $\sigma(\cdot)$ is the sigmoid function.

## 5 Experiments

### 5.1 Implementation Details

**Training Setup** To validate the effectiveness of the proposed methods, we use the pair preference datasets from BPO (Pi et al., 2024). This dataset contains 180k pairwise preference data, where the negative responses are generated by the Image-Weakened prompting and LLM Error Injection. For training parameters, we train the model for 1 epoch and set the $\beta$ to 0.2. For LLaVA V1.5, we follow previous work (Pi et al., 2024) to adopt the LoRA (Hu et al., 2021) training with rank 32 and LoRA alpha 256. The learning rate is set to 2e-6. For LLaVA-OneVision, we use the recommended official training script to perform full fine-tuning where the learning rate is 5e-7. The training takes about 7 hours for LLaVA V1.5 7B models and 22 hours for LLaVA-OneVision 7B. For the analysis section, we preserve the top and bottom 20% scores as high-confidence CLIP-based predictions.

**Choice of VLMs** For the choice of VLMs, we employ the CLIP (Radford et al., 2021) ViTL/14@336. For the preferred and negative text responses in each data sample, we encode the longest possible text from

Table 1: **Results on hallucination and comprehensive benchmarks.** For each column, the ↑ symbol indicates that higher values are better, while the ↓ symbol indicates that lower values are better. The *AMB. Gen.* and *AMB. Dis.* refer to the generative and discriminative components of the AMBER benchmark. The ∗ symbol denotes the evaluation results of the official model weights from BPO (Pi et al., 2024). We clarify that ‡ focuses on utilizing CLIP knowledge to construct preference data, whereas we leverage CLIP to address the overfitting issue in standard DPO training. The mDPO† provides an alternative approach to improving multimodal preference optimization by addressing the over-prioritization of language preference, which is orthogonal to the DA-DPO approach.

| | AMB.$^G$ | | | | AMB.$^D$ | ObjHal | | MMHal | | POPE | LLaVA$^W$ | GQA | Seed$^I$ | MME$^P$ | MME$^C$ |
|---|---|---|---|---|---|---|---|---|---|---|---|---|---|---|---|
| | C$_s$↓ | Cov. ↑ | Hal. ↓ | Cog. ↓ | F1 ↑ | C$_s$↓ | C$_i$↓ | Score ↑ | HalRate ↓ | F1 ↑ | Score ↑ | Acc↑ | Acc↑ | Score↑ | Score↑ |
| *Reference Only (Not directly comparable)* | | | | | | | | | | | | | | | |
| CogVLM | 5.6 | 57.2 | 23.6 | 1.3 | - | - | - | - | - | - | - | - | - | - | - |
| mPLUG-Owl2 | 10.6 | 52.0 | 39.9 | 4.5 | - | - | - | - | - | - | 56.1 | - | - | - | - |
| InstructBLIP | 8.8 | 52.2 | 38.2 | 4.4 | - | - | - | - | - | - | 58.2 | - | 58.8 | 1212.8 | - |
| Qwen-VL | 5.5 | 49.4 | 23.6 | 1.9 | - | 36.0 | 21.3 | 2.89 | 0.43 | - | - | 59.3 | - | 1487.6 | 360.7 |
| GPT-4V | 4.6 | 67.1 | 30.7 | 2.6 | - | 13.6 | 7.3 | 3.49 | 0.28 | - | - | - | - | - | - |
| LLaVA-v1.5-7B | 7.8 | 51.0 | 36.4 | 4.2 | 74.7 | 54.7 | 15.9 | 2.19 | 0.57 | 85.8 | 63.8 | 62.0 | 66.1 | 1510.7 | 307.5 |
| + HA-DPO | 7.2 | 33.6 | 19.7 | 2.6 | - | 39.9 | 19.9 | 1.97 | 0.60 | - | - | - | - | - | - |
| + CLIP-DPO‡ | 3.7 | 47.8 | 16.6 | 1.3 | - | - | - | - | - | 85.8 | - | - | - | 1468.7 | - |
| + mDPO† | 4.4 | 52.4 | 24.5 | 2.4 | - | 35.7 | 9.8 | 2.39 | 0.54 | - | - | - | - | - | - |
| *Main Results* | | | | | | | | | | | | | | | |
| LLaVA-v1.5-7B | 7.8 | 51.0 | 36.4 | 4.2 | 74.7 | 54.7 | 15.9 | 2.19 | 0.57 | 85.8 | 63.8 | **62.0** | **66.1** | **1510.7** | 307.5 |
| + DPO* | 5.5 | **58.4** | 35.7 | **2.0** | 83.9 | 43.3 | 10.0 | **2.24** | 0.53 | 84.3 | 70.5 | 45.3 | 57.8 | 1409.4 | 315.0 |
| + DA-DPO | **4.3** | 57.4 | **28.0** | 2.1 | **85.6** | 39.7 | 9.9 | 2.22 | **0.50** | **85.9** | **75.4** | 59.8 | 64.8 | 1406.6 | **323.2** |
| LLaVA-v1.5-13B | 7.1 | 52.4 | 33.9 | 3.8 | **88.6** | 56.3 | 15.8 | 2.22 | 0.57 | **86.9** | 71.2 | **63.0** | **67.5** | **1496.7** | 308.9 |
| + DPO | 6.6 | **61.9** | 45.7 | 2.5 | 85.5 | 39.0 | 8.8 | 2.05 | 0.55 | 86.0 | 74.1 | 60.9 | 53.5 | 1450.6 | 276.4 |
| + DA-DPO | **5.1** | 59.0 | **32.2** | **1.9** | 87.3 | **37.0** | **8.3** | **2.39** | **0.48** | 85.6 | **74.5** | 61.5 | 61.4 | 1474.2 | 276.4 |
| LLaVA-OV-7B | 8.4 | **74.8** | 70.4 | 9.8 | 90.6 | 41.3 | 8.1 | 2.76 | 0.37 | **87.2** | **80.0** | 59.6 | 75.8 | 1580.4 | 406.4 |
| + DPO | 7.7 | 66.5 | 59.0 | 4.3 | **91.9** | 34.3 | 8.6 | 2.61 | 0.38 | 86.1 | 72.9 | 55.0 | 73.7 | 1545.4 | 364.6 |
| + DA-DPO | **7.0** | 64.1 | **51.8** | **3.7** | 91.7 | **28.0** | **8.0** | **2.78** | **0.30** | 86.0 | 73.7 | 59.2 | 75.6 | 1551.4 | 381.8 |

the start and truncate the rest of the text. For MLLM, we use the LLaVA v1.5 7B (Liu et al., 2023a) to compute the probability of responses given the image and question. We also provide an ablation study on the choice of these models in Section 5.5.

## 5.2 Evaluation Benchmarks

To comprehensively evaluate the impact of preference optimization on MLLMs, we select two types of benchmarks. Hallucination benchmarks measure the model's ability to reduce factual errors, which is the primary goal of multimodal preference alignment. Comprehensive benchmarks assess general multimodal capabilities, ensuring that improvements in hallucination do not come at the cost of overall performance.

**Hallucination Evaluation.** Following previous works (Pi et al., 2024; Wang et al., 2024a; Ouali et al., 2024), we comprehensively evaluate the DA-DPO on various hallucination benchmarks such as AMBER (Wang et al., 2023a), MMHalBench (Sun et al., 2023b), Object HalBench (Rohrbach et al., 2018), and POPE (Li et al., 2023e). 1) AMBER provides a multidimensional framework suitable for assessing both generative and discriminative tasks. 2) MMHalBench is a question-answering benchmark with eight question types and 12 object topics. We follow the official evaluation scripts with GPT-4. 3) Object HalBench is a standard benchmark for assessing object hallucination, and we follow the settings in (Wang et al., 2024a).

Table 2: **The impact of estimation from different VLMs.** We report the performance of the model trained with different strategies on hallucination benchmarks such as the AMBER benchmark and the comprehensive benchmark SeedBench. The row with the blue color indicates the adopted strategy.

| | VLMs | | AMB.$^G$ | | | | AMB.$^D$ | Seed$^I$ |
|---|---|---|---|---|---|---|---|---|
| | CLIP | MLLM | $C_s$ ↓ | Cov. ↑ | Hal. ↓ | Cog. ↓ | F1 ↑ | Score ↑ |
| DPO | × | × | 5.5 | 58.4 | 35.7 | 2.0 | 83.9 | 57.8 |
| | ✓ | × | 4.7 | **58.0** | 31.9 | 2.0 | 85.2 | 63.8 |
| Ours | × | ✓ | 4.6 | 57.6 | 29.9 | 2.3 | 85.3 | 64.3 |
| | ✓ | ✓ | **4.3** | 57.4 | **28.0** | **2.1** | **85.6** | **64.8** |

Table 3: **Results of LLaVA v1.5 7B trained with VLFeedback and LLaVA-RLHF dataset.** The VLFeedback is an automatically constructed dataset by collecting responses from multiple VLMs and filtering the results with GPT4-V. The LLaVA-RLHF is a human-annotated dataset.

| | AMB.$^G$ | | | | AMB.$^D$ | ObjHal | | MMHal | | POPE | LLaVA$^W$ | GQA | Seed$^I$ | MME$^P$ | MME$^C$ |
|---|---|---|---|---|---|---|---|---|---|---|---|---|---|---|---|
| | $C_s$ ↓ | Cov. ↑ | Hal. ↓ | Cog. ↓ | F1 ↑ | $C_s$ ↓ | $C_i$ ↓ | Score ↑ | HalRate ↓ | F1 ↑ | Score ↑ | Acc↑ | Acc↑ | Score↑ | Score↑ |
| LLaVA-v1.5-7B | 7.8 | 51.0 | 36.4 | 4.2 | 74.7 | 54.7 | 15.9 | 2.19 | 0.57 | 85.8 | 63.8 | 62.0 | 66.1 | 1510.7 | 307.5 |
| | | | | | | *VLFeedback* | | | | | | | | | |
| + DPO | 6.5 | **55.1** | 34.5 | **2.3** | 84.6 | 49.0 | 13.0 | 2.19 | 0.65 | 84.6 | 72.1 | 59.8 | 63.5 | 1368.5 | 294.2 |
| + DA-DPO | **5.6** | 53.0 | **29.7** | 2.7 | **85.8** | **48.3** | **12.8** | **2.23** | **0.53** | 84.2 | **73.0** | **60.3** | **65.0** | **1422.7** | **297.1** |
| | | | | | | *LLaVA-RLHF* | | | | | | | | | |
| + DPO | 7.3 | **50.8** | 33.7 | 3.9 | 86.6 | 58.3 | 16.9 | 2.10 | **0.57** | 84.4 | 68.7 | 60.7 | 64.5 | 1415.9 | **343.9** |
| + DA-DPO | **5.8** | 50.7 | **27.6** | **3.0** | 86.6 | **48.0** | **13.8** | **2.02** | 0.59 | **84.5** | **71.3** | **60.9** | **64.8** | **1464.1** | 301.4 |

4) POPE utilizes a polling-based query to evaluate the model's hallucination. We report the average F1 score of three kinds of questions in POPE.

**Comprehensive Evaluation.** For evaluating MLLM helpfulness, we use: 1) LLaVA-Bench (Liu et al., 2023b), a real-world benchmark with 60 tasks assessing LLaVA's visual instruction-following and question-answering abilities. We use official scripts to compute scores with GPT-4. 2) Seedbench (Li et al., 2023a), which consists of 14k multiple-choice VQA samples to evaluate the comprehensive ability of MLLMs. 3) MME (Fu et al., 2023) which measures both perception and cognition abilities using yes/no questions. 4) GQA Hudson & Manning (2019) evaluates real-world visual reasoning and compositional question-answering abilities in an open-ended answer generation format.

### 5.3 Baselines

The proposed DA-DPO framework is designed to improve pairwise preference optimization by introducing the pretrained VLMs in a cost-efficient manner. We mainly compare DA-DPO with standard DPO (Rafailov et al., 2024) under three MLLMs, LLaVA V1.5 7B/13B (Liu et al., 2023a) and LLaVA-OneVision 7B (Li et al., 2024). Additionally, we provide results from other multimodal LLMs, such as GPT4-V (OpenAI, 2023), CogVLM (Wang et al., 2023b), mPLUG-Owl2 (Ye et al., 2023), InstructBLIP (Dai et al., 2023), Qwen-VL (Bai et al., 2023), HA-DPO (Zhao et al., 2023), CLIP-DPO (Ouali et al., 2024), mDPO (Wang et al., 2024a) for reference.

### 5.4 Results Analysis

**Hallucination Benchmarks** To demonstrate the effectiveness of the proposed methods in reducing hallucinations, we present evaluation results on various hallucination benchmarks in Table 1. We observe that the DA-DPO improves model performance on most benchmarks compared to DPO, such as the *HalRate* in AMBER generative decrease from 35.7 to 28.0 when training with LLaVA v1.5 7B. Moreover, the performance

Table 4: **Choices of VLMs.** The *ViTL* indicate the CLIP ViTL/14@336 and *EVA 8B* is the EVA-CLIP 8B. The *LLaVA 7B* is the LLaVA v1.5 7B and the *OV 7B* is the LLaVA-Onevision 7B.

| Model Choice | | AMB.$^G$ | | | | AMB.$^D$ | Seed$^I$ |
|---|---|---|---|---|---|---|---|
| CLIP | MLLM | $C_s$ ↓ | Cov. ↑ | Hal. ↓ | Cog. ↓ | F1 ↑ | Score ↑ |
| ViTL | LLaVA 7B | **4.3** | 57.4 | 28.0 | 2.1 | 85.6 | 64.8 |
| EVA 8B | LLaVA 7B | 4.9 | 57.1 | 31.8 | 2.6 | 83.4 | 56.2 |
| ViTL | OV 7B | 4.6 | **57.7** | 31.4 | 2.0 | 85.7 | 65.2 |
| EVA 8B | OV 7B | **4.3** | 55.6 | **25.5** | 2.0 | **86.0** | **65.5** |

Table 5: **Comparison of DA-DPO with direct filtering baselines (10%, 25%, 50% easy samples removed).** Metrics with ↑ indicate higher is better; with ↓ indicate lower is better.

| Model | AMBER$^G$ | | | | AMBER$^D$ | POPE | MMHal | | ObjHall | | LLaVA$^W$ | GQA | Seed | MME | |
|---|---|---|---|---|---|---|---|---|---|---|---|---|---|---|---|
| | $C_s$↓ | Cov.↑ | Hal.↓ | Cog.↓ | F1↑ | F1↑ | Score↑ | HalRate↓ | $C_s$↓ | $C_i$↓ | Score↑ | Acc↑ | Acc↑ | P↑ | C↑ |
| DPO | 5.5 | 58.4 | 35.7 | **2.0** | 83.9 | 84.3 | 2.23 | 0.53 | 43.3 | 10.0 | 70.48 | 45.3 | 57.7 | **1409.4** | 315.0 |
| Filter 10% | 5.6 | **63.2** | 39.6 | 2.6 | 85.9 | 84.9 | 2.22 | 0.51 | 39.0 | 9.3 | 72.49 | 53.4 | 50.3 | 1365.1 | 300.0 |
| Filter 25% | 5.3 | 61.3 | 37.8 | 2.3 | 85.8 | 84.3 | **2.29** | **0.50** | 38.3 | **9.0** | 67.04 | 53.3 | 51.1 | 1318.6 | **330.3** |
| Filter 50% | 5.3 | 62.2 | 38.3 | 2.0 | **86.0** | 85.5 | 2.03 | 0.56 | 42.7 | 10.2 | 71.21 | **56.3** | 52.2 | 1359.7 | 306.7 |
| DA-DPO | **4.3** | 57.4 | **28.0** | 2.1 | 85.6 | **85.9** | 2.22 | **0.50** | **37.7** | 9.8 | **75.38** | **59.8** | **64.8** | 1406.5 | 323.2 |

of three MLLMs is improved significantly on the Object Hallucination benchmark, which demonstrates that the DA-DPO greatly reduces the hallucination in the caption.

**Comprehensive Benchmarks**   We evaluate the proposed methods on five comprehensive benchmarks. The results, shown in Table 1, indicate that while preference optimization reduces hallucination, performance on general abilities suffers. However, DA-DPO mitigates this degradation on most benchmarks when compared to standard DPO. Furthermore, DA-DPO significantly enhances conversational ability, with performance on the LLaVA-Bench reaching 75.4, compared to 70.5 for DPO using LLaVA v1.5 7B.

## 5.5   Ablation Study

**Impact of the Difficulty-aware Training**   To better understand the proposed methods, we conduct an ablation study to assess the effectiveness of each design. As shown in Table 2, we begin with the standard Direct Preference Optimization (DPO) (Rafailov et al., 2024). We then add the reward score from the CLIP to control sample weight during training, followed by results for DPO with only the MLLM's reward score. Finally, we combine both reward scores using the adaptive fusion strategy. The results show that using either CLIP or MLLM's reward score improves both *HalRate* and *CHAIR$_s$*, with the combination of both achieving the best performance, highlighting the necessity of each framework design.

**Influence of Preference Datasets**   To demonstrate that the proposed methods mitigate data bias in different types of multimodal pairwise preference data, we use VLFeedBack (Li et al., 2023d), which consists of 80k responses generated by MLLMs with varying levels of ability, and rated by GPT4-V (OpenAI, 2023). This dataset is automatically generated through a different approach compared to BPO (Pi et al., 2024), and it covers the mainstream data generation pipeline for multimodal preference data. We follow the settings of mDPO (Wang et al., 2024a) to select 10k preference data samples. Moreover, we trained on LLaVA-RLHF (Sun et al., 2023b), the most widely used human-annotated multimodal preference dataset with 10k samples. As shown in Table 3, DA-DPO outperforms DPO on most benchmarks, demonstrating that overfitting is a common issue in multimodal preference data, and DA-DPO effectively alleviates this problem in a cost-efficient manner.

**Choices of the VLMs**   We present an ablation study on the selection of VLMs. These models estimate the difficulty of pairwise preference data and guide difficulty-aware preference optimization during training

Table 6: **Performance across difficulty buckets estimated using CLIP-based and MLLM-based proxies.** We fine-tune LLaVA-1.5-7B using vanilla DPO, where the training data in each bucket is sampled from the BPO dataset based on the estimated reward gap. The *Top 0–25%* bucket corresponds to samples with the smallest reward gaps.

| Bucket | AMB.[G] | | | | AMB.[D] | ObjHal | | POPE |
|---|---|---|---|---|---|---|---|---|
| | $C_s\downarrow$ | Cov.$\uparrow$ | Hal.$\downarrow$ | Cog.$\downarrow$ | F1$\uparrow$ | $C_s\downarrow$ | $C_i\downarrow$ | F1$\uparrow$ |
| *CLIP Estimation* | | | | | | | | |
| Top 0–25% | 5.3 | 62.0 | 37.6 | 2.2 | **86.8** | **41.3** | 10.8 | 85.6 |
| Top 25–50% | **5.1** | **62.7** | **37.5** | **2.0** | 86.2 | 43.0 | **9.7** | **85.8** |
| Top 50–75% | 5.2 | 61.1 | 37.9 | 2.4 | 85.7 | 51.0 | 10.8 | 85.8 |
| Top 75–100% | 5.4 | 55.0 | 38.9 | 2.5 | 85.6 | 52.7 | 12.9 | 85.7 |
| *MLLM Estimation* | | | | | | | | |
| Top 0–25% | 4.9 | **59.1** | 35.4 | 2.5 | 85.6 | 46.7 | 11.8 | 85.7 |
| Top 25–50% | **4.6** | 53.9 | **32.3** | **1.9** | **86.6** | **42.3** | **9.0** | 85.8 |
| Top 50–75% | 5.4 | 55.5 | 30.5 | 2.0 | 84.6 | 48.3 | 11.4 | **86.5** |
| Top 75–100% | 5.5 | 54.3 | 32.4 | 2.2 | 84.8 | 46.3 | 10.8 | 86.3 |

in the proposed framework. We experiment with two CLIP models of varying parameter scales: CLIP ViT-L/14@336 (Radford et al., 2021) and EVA-CLIP 8B (Sun et al., 2023a). For MLLMs, we use LLaVA v1.5 7B (Liu et al., 2023a) and LLaVA-OneVision 7B (Li et al., 2024). As shown in Table 4, we observe that performance remains similar across VLMs with different capabilities, which is attributed to the Gaussian normalization in Eq. (10) and Eq. (6). This normalization ensures that the VLMs only provide a ranking of preference data, making our proposed framework robust to variations in the choice of VLMs.

**Comparison with Direct Filtering Baseline**  To further validate the effectiveness of our sample re-weighting strategy, we compare DA-DPO against a baseline that directly filters out *easy samples* from the training data. Specifically, we remove 10%, 25%, and 50% of the easy samples from the BPO dataset based on our difficulty metric and train a model using the DPO algorithm. Table 5 summarizes the performance of DA-DPO and the direct filtering baselines across a wide range of benchmarks. We observe that DA-DPO consistently outperforms the filtering-based approaches on most metrics, especially on hallucination-related benchmarks such as POPE, MMHal, and Object Hallucination. Notably, the performance of the direct filtering baseline fluctuates across different filtering ratios, with no consistent improvement as more easy samples are removed.

We attribute the limited effectiveness of direct filtering to its inherent trade-off: while it removes truly uninformative samples, it also discards a portion of valuable training data, thereby reducing the diversity and coverage of preference information. In contrast, DA-DPO addresses this issue by softly down-weighting easy samples instead of hard filtering, preserving data diversity while emphasizing informative examples. This allows DA-DPO to maintain robustness across benchmarks with varying difficulty and annotation styles.

**Estimated Reward Gap and Model Hallucination**  To examine the relationship between the reward gap and hallucination behavior, we design a controlled validation experiment. Specifically, we utilize CLIP-based and MLLM-based proxies to estimate the reward gaps of training samples and evenly divide the data into four buckets according to the estimated gap range. We then train a vanilla DPO model on each bucket independently to analyze how sample difficulty affects preference optimization. As shown in Table 6, we observe that the largest reduction in hallucination occurs for samples in the 25–50% bucket, while the extent of hallucination reduction gradually decreases as the reward gap increases. Interestingly, the bucket with the smallest reward gap (0–25%) does not yield the most significant hallucination improvement. We attribute this to noisy or confusing samples within this subset during data construction—such as cases where the "rejected" answer is actually semantically correct—which may interfere with effective model optimization.

# 6 Related Works

## 6.1 Vision-Language Models

Vision-Language Models (VLMs) (Radford et al., 2021; Jia et al., 2021; Li et al., 2022) have substantially advanced multimodal understanding, achieving strong performance across a wide range of downstream tasks (Guo et al., 2022; Ning et al., 2023). Contrastive VLMs, such as CLIP (Radford et al., 2021), align images and text via large-scale contrastive objectives and demonstrate impressive zero-shot transfer on recognition tasks (Nukrai et al., 2022; Qiu et al., 2024; Liao et al., 2022; Ning et al., 2023; Yao et al., 2022; Peebles & Xie, 2022). With the rise of large language models (LLMs), subsequent works integrate LLMs with visual encoders to enhance image-conditioned text generation (Li et al., 2023b; Dai et al., 2023; Liu et al., 2023b; Zhu et al., 2023; Chen et al., 2023; Lin et al., 2023), enabling instruction following and open-ended reasoning. More recent efforts (Liu et al., 2023a; Li et al., 2024; Gao et al., 2024; Bai et al., 2025) further improve visual capabilities by leveraging higher-resolution inputs, multi-image contexts, and even video sequences. Despite these advances, VLMs remain prone to hallucinations when aligning visual evidence with textual responses, motivating preference-based optimization approaches to improve faithfulness.

## 6.2 Preference Alignment in Large Language Models

The alignment problem (Leike et al., 2018) aims to ensure that agent behaviors are consistent with human intentions. Early approaches leveraged Reinforcement Learning from Human Feedback (RLHF) (Bai et al., 2022a;b; Glaese et al., 2022; Nakano et al., 2021; Ouyang et al., 2022; Scheurer et al., 2023; Stiennon et al., 2020; Wu et al., 2021; Ziegler et al., 2019), where policy optimization methods such as PPO (Schulman et al., 2017) were used to maximize human-labeled rewards. More recently, Direct Preference Optimization (DPO) (Rafailov et al., 2024) reformulates alignment as a direct optimization problem over offline preference data, avoiding the need for reinforcement learning. Extensions such as Gibbs-DPO (Xiong et al., 2023) enable online preference optimization, while $\beta$-DPO (Wu et al., 2024) addresses sensitivity to the temperature parameter $\beta$ by introducing dynamic calibration with a reward model. However, such approaches are less effective in multimodal domains, where reward models are vulnerable to reward hacking (Sun et al., 2023b). Other works explore alternative strategies, such as self-rewarding mechanisms or data reweighting (Wang et al., 2024b; Zhou et al., 2024a), to mitigate issues like overconfident labeling and distributional bias in preference data.

## 6.3 Preference Alignment in Multimodal Models

Recent efforts have extended preference alignment from language-only models to the multimodal domain. A major line of work focuses on constructing multimodal preference datasets. (Sun et al., 2023b; Yu et al., 2024a) collect human annotations, while others rely on powerful multimodal models such as GPT-4V (Li et al., 2023c; Yu et al., 2024b; Zhou et al., 2024d; Yang et al., 2025) to generate preference signals. However, both human annotation and large model inference incur prohibitive costs, limiting scalability. To address this, alternative approaches (Deng et al., 2024; Pi et al., 2024) explore automatic or self-training methods for preference data generation. These works synthesize dis-preferred responses from corrupted images or misleading prompts, enabling the model to learn preferences without external supervision. Another direction (Ouali et al., 2024) leverages CLIP to score diverse candidate responses, ranking preferred versus dis-preferred outputs using image–text similarity. From the perspective of training objectives, most multimodal alignment methods adopt the standard DPO objective (Li et al., 2023c; Zhao et al., 2023; Zhou et al., 2024b) to optimize preferences on paired data. Other approaches employ reinforcement learning (Sun et al., 2023b) or contrastive learning (Sarkar et al., 2024; Jiang et al., 2024) to improve alignment. To further reduce overfitting to language-only signals, (Ouali et al., 2024) extends DPO by jointly optimizing both textual and visual preferences. Despite these advances, multimodal DPO remains vulnerable to overfitting, especially when trained on imbalanced preference data. In this work, we introduce a general difficulty-aware training framework that explicitly accounts for sample difficulty, thereby mitigating overfitting and improving robustness in multimodal preference optimization.

# 7 Conclusion

In this work, we present an empirical analysis of the overfitting issue in multimodal preference optimization, which often stems from imbalanced data distributions. To address this, we introduce DA-DPO, a cost-efficient framework consisting of difficulty estimation and difficulty-aware training. Our method leverages pretrained contrastive and generative VLMs to estimate sample difficulty in a training-free manner, and uses these estimates to adaptively reweight data—emphasizing harder samples while preventing overfitting to easier ones. Experiments across hallucination and general-purpose benchmarks demonstrate that this paradigm effectively improves multimodal preference optimization.

**Limitations** Despite these promising results, our framework relies on the assumption that pretrained VLMs provide reliable evaluations of preference data. Although the adaptive voting strategy shows robustness on existing datasets, its generalizability to domains that differ substantially from the pretraining objectives of these VLMs remains uncertain. Future work may explore integrating domain-adaptive or self-improving mechanisms to further enhance robustness.

# 8 Acknowledgements

This work was supported by NSFC 62350610269, Shanghai Frontiers Science Center of Human-centered Artificial Intelligence, and MoE Key Lab of Intelligent Perception and Human-Machine Collaboration (ShanghaiTech University). This work was also supported by HPC platform of ShanghaiTech University.

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

Table 7: **Fusion Ratio Ablation.** The *Fusion Ratio* indicate the weight for the $\hat{c}_g$ and $\hat{m}_g$ in Equation 11. The row with blue color is the weight of adaptive score fusion weight.

| Fusion Ratio | | AMB.$^G$ | | | | AMB.$^D$ | Seed$^I$ |
|---|---|---|---|---|---|---|---|
| CLIP | MLLM | C$_s$ ↓ | Cov. ↑ | Hal. ↓ | Cog. ↓ | F1 ↑ | Score ↑ |
| 20% | 80% | 4.9 | 57.7 | 30.9 | **2.0** | 85.4 | 63.2 |
| 40% | 60% | **4.3** | 57.5 | 28.3 | **2.0** | 85.6 | 64.0 |
| 53% | 47% | **4.3** | 57.4 | **28.0** | 2.1 | **85.6** | **64.8** |
| 60% | 40% | 4.5 | 57.6 | 28.4 | 2.1 | 85.3 | 63.8 |
| 80% | 20% | 4.6 | **58.3** | 28.6 | **2.0** | 85.5 | 62.8 |

Table 8: **Performance with different $\beta$ for DA-DPO and DPO.** We provide the results of DPO and our difficulty-aware DPO trained in the BPO dataset. We mark the reported hyperparameter with blue. For DA-DPO, we chose the best performance $\beta$, which is 0.2. For DPO, we follow previous work reports that $\beta$ equals to 0.1.

| | AMB.$^G$ | | | | AMB.$^D$ | ObjHal | | POPE | GQA | Seed$^I$ | MME$^P$ | MME$^C$ |
|---|---|---|---|---|---|---|---|---|---|---|---|---|
| | C$_s$ ↓ | Cov. ↑ | Hal. ↓ | Cog. ↓ | F1 ↑ | C$_s$ ↓ | C$_i$ ↓ | F1 ↑ | Score ↑ | Acc ↑ | Score ↑ | Score ↑ |
| | | | | | *DA-DPO* | | | | | | | |
| $\beta = 0.1$ | 4.6 | **59.0** | 32.5 | **2.0** | 85.4 | 40.4 | 10.2 | 85.6 | 58.6 | 64.2 | 1398.2 | 312.8 |
| $\beta = 0.2$ | **4.3** | 57.4 | 28.0 | 2.1 | 85.6 | 39.7 | 9.9 | **85.9** | **59.8** | **64.8** | 1406.6 | **323.2** |
| $\beta = 0.3$ | 4.7 | 53.6 | **25.8** | 2.4 | **86.3** | **38.6** | **9.5** | 84.8 | 56.4 | 58.4 | **1414.2** | 315.6 |
| $\beta = 0.4$ | 4.9 | 54.2 | 28.4 | 2.6 | 86.0 | 41.6 | 10.4 | 85.2 | 58.4 | 62.2 | 1378.2 | 306.9 |
| | | | | | *DPO* | | | | | | | |
| $\beta = 0.1$ | 5.5 | 58.4 | 35.7 | **2.0** | 83.9 | **43.3** | **10.0** | 84.3 | 45.3 | **57.7** | **1409.4** | **315.0** |
| $\beta = 0.2$ | **4.8** | **58.5** | 30.0 | **2.0** | 83.4 | 44.2 | 10.2 | 83.8 | 58.5 | 47.9 | 1369.1 | 299.6 |
| $\beta = 0.3$ | 4.9 | 56.6 | **29.4** | **2.0** | 84.4 | 45.3 | 10.8 | 84.3 | 56.8 | 54.7 | 1387.1 | 310.7 |
| $\beta = 0.4$ | 4.9 | 56.5 | 30.8 | **2.0** | **84.5** | 45.3 | 12.1 | **85.2** | **58.9** | 55.4 | 1402.4 | 312.3 |

## A  Influence of Adaptive Score Fusion

The proposed framework integrates two types of pretrained VLMs to assess the difficulty of pairwise preference data. We design an adaptive score fusion mechanism to determine the weight of each reward model without requiring hyperparameter selection from the data perspective. As shown in Table 7, we present the results for multiple fixed fusion scores. We observe that the adaptive score fusion achieves the best performance.

## B  Sensitivity of $\beta$

We propose a difficulty-aware preference optimization strategy that aims to alleviate the overfitting-to-easy-sample issue during alignment training. To achieve this, we dynamically calibrate the $\beta$, which is described in the main paper. However, our method affects the scale of the $\beta$ in the DPO objective. As shown in Table 8, we provide the ablation of the $\beta$ between the DPO and DA-DPO. From the results, we observe that varying $\beta$ leads to mixed performance across different benchmarks for both DPO and DA-DPO. Nevertheless, the proposed DA-DPO consistently achieves better overall performance compared to vanilla DPO, indicating that our difficulty-aware calibration effectively enhances robustness while mitigating hallucinations.

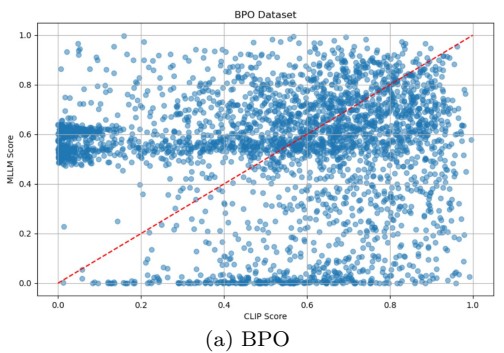 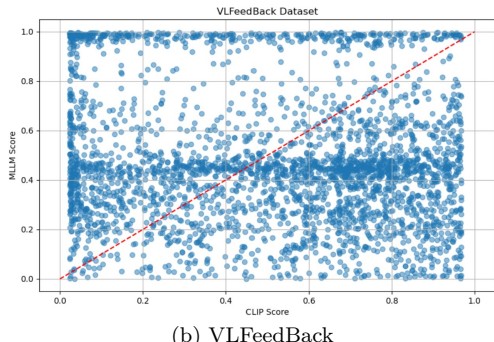

(a) BPO                    (b) VLFeedBack

Figure 5: **The visualization correlation between pretrained VLMs.** We present the normalized MLLM and CLIP scores for the same preference data sample in three datasets, BPO and VLFeedBack. The x-axis is the CLIP score and the y-axis is the MLLM score. The red line indicates the MLLM and CLIP's predictions are the same.

Table 9: Classification accuracy of fusion scores on the BPO and VLFeedback datasets.

| Dataset | RM Type | VQA | Caption | Text VQA | Overall |
|---|---|---|---|---|---|
| BPO | CLIP | 0.5877 | 0.8947 | 0.9827 | **0.7732** |
| BPO | MLLM | 0.8478 | 0.5053 | 0.6282 | **0.6415** |
| VLFeedback | CLIP | 0.5827 | 0.6352 | – | **0.5897** |
| VLFeedback | MLLM | 0.8104 | 0.5193 | – | **0.7715** |

## C  Pretrained VLMs Correlation

In our proposed DA-DPO framework, we utilize the CLIP and the MLLM to evaluate the difficulty of preference data from different perspectives. To validate this claim, we provide the score of difficulty correlation between the two VLMs. As shown in Figure 5, the scores from the two models exhibit no significant positive correlation, as evidenced by their weak correlation coefficient. This indicates that the two models capture different aspects of the response evaluation, and their scoring patterns do not align consistently across the datasets.

## D  Fusion Score Classification Accuracy

To provide a comprehensive understanding of the score fusion process, we supplement the main text with detailed numerical results and evaluation metrics. Score fusion is conducted at the dataset level, where distinct fusion scores are employed for the BPO and VLFeedback datasets.

**Evaluation Metric.** We compute classification accuracy on pairwise preference data using the following criterion: for each sample, the VLM assigns a score to both the chosen and rejected answers. A sample is considered correctly classified if the chosen answer receives a higher score than the rejected one.

**Results.** Table 9 reports classification accuracy for each category (VQA, Caption, and Text VQA, where applicable) and the overall accuracy on both datasets. We evaluate two types of VLMs: a vision encoder (CLIP) and a multimodal large language model (MLLM). Specifically, we use EVA-CLIP 8B as the CLIP model and LLaVA-1.5 7B as the MLLM.

Based on the overall accuracy, we designate the CLIP-based classification score (`cls_c`) and MLLM-based classification score (`cls_m`) as follows: For the BPO dataset, `cls_c` = 0.7732 and `cls_m` = 0.6415; for the VLFeedback dataset, `cls_c` = 0.5897 and `cls_m` = 0.7715. These results demonstrate the complementary strengths of CLIP and MLLM-based reward estimations across different categories.

Table 10: **Compute and Time Estimation for Reward Model and Proxies.** FLOPs are shown in scientific notation, and Wall-clock hours are estimated on an NVIDIA A100 GPU. The data for the reward model training is set to 180k.

| Task | FLOPs | Wall-clock hours (A100) |
|---|---|---|
| *Traditional Approach* | | |
| Training Reward Model | $2.6 \times 10^{19}$ | 241.9 |
| Reward Model Estimation | $5.1 \times 10^{18}$ | 40 |
| *DA-DPO* | | |
| CLIP Estimation | $1.4 \times 10^{16}$ | 0.7 |
| MLLM Estimation | $5.1 \times 10^{18}$ | 40 |

Table 11: **Ablation of data difficulty normalization.** We report the performance of different normalization strategies trained with LLaVA-v1.5 7B on the BPO dataset.

| | AMB.$^G$ | | | | AMB.$^D$ | ObjHal | | POPE | GQA | Seed$^I$ | MME$^P$ | MME$^C$ |
|---|---|---|---|---|---|---|---|---|---|---|---|---|
| | $C_s \downarrow$ | Cov. $\uparrow$ | Hal. $\downarrow$ | Cog. $\downarrow$ | F1 $\uparrow$ | $C_s \downarrow$ | $C_i \downarrow$ | F1 $\uparrow$ | Score $\uparrow$ | Acc $\uparrow$ | Score $\uparrow$ | Score $\uparrow$ |
| DPO | 5.5 | **58.4** | 35.7 | 2.0 | 83.9 | 43.3 | 10.0 | 85.6 | 45.3 | 57.7 | **1409.4** | 315.0 |
| Ranked-based | 4.8 | 57.2 | 31.3 | **1.9** | 85.7 | 42.7 | 10.7 | **86.1** | 59.5 | 62.1 | 1404.8 | 302.5 |
| Length-controlled | 4.7 | 57.0 | 29.3 | **1.9** | **85.9** | **37.7** | **9.6** | 85.9 | **60.0** | 64.1 | 1402.6 | 316.0 |
| Gaussian Normalization | **4.3** | 57.4 | **28.0** | 2.1 | 85.6 | 39.7 | 9.9 | 85.9 | 59.8 | **64.8** | 1406.6 | **323.2** |

## E Efficiency Comparison

To validate the efficiency advantage of our proposed framework, we compare the computational cost in terms of FLOPs and wall-clock time between traditional reward model training and evaluation, and our estimation-based approach. As summarized in Table 10, training a full reward model requires over 6 times the wall-clock time of MLLM-based estimation and approximately 345 times that of CLIP-based estimation. These results clearly demonstrate that our method achieves significantly higher efficiency compared to conventional approaches.

## F Difficulty Estimation Normalization Ablation

To evaluate the impact of different normalization strategies on data difficulty estimation, we conducted experiments with two alternative approaches. The first strategy, *Ranked-based*, normalizes scores by ranking the data and projecting the values linearly into the $[0, 1]$ range, instead of using Gaussian normalization. The second strategy, *Length-controlled*, is applied during the MLLM generative estimation: the sum of log probabilities in Equations 7 and 8 is divided by the response token length to mitigate length bias. As shown in Table 11, all three normalization strategies achieve performance improvements over vanilla DPO. We also observe that the Ranked-based approach slightly underperforms the other two strategies, suggesting that preserving the original distribution of estimated scores is beneficial for effective difficulty estimation.

## G Experiments with more Model Variants

To validate the effectiveness of our proposed method beyond the LLaVA series, we conduct a small-scale experiment on Qwen2.5-VL 3B. We use a subset of 50k samples from the BPO dataset to train both the baseline DPO and our proposed DA-DPO. The results are presented in Table 12. We observe that DA-DPO achieves better reductions in hallucination metrics compared to DPO, while retaining performance on comprehensive benchmarks. This demonstrates that DA-DPO is a general framework, not limited to LLaVA-series models.

Table 12: **Comparison of hallucination and general benchmarks across methods.** Results are reported on Qwen2.5-VL 3B and models trained with DPO and DA-DPO. The proposed DA-DPO consistently reduces hallucination metrics while maintaining or improving general ability.

| | AMB.$^G$ | | | | AMB.$^D$ | ObjHal | | POPE | GQA | Seed$^I$ | MME$^P$ | MME$^C$ |
| | $C_s$ ↓ | Cov. ↑ | Hal. ↓ | Cog. ↓ | F1 ↑ | $C_s$ ↓ | $C_i$ ↓ | F1 ↑ | Score ↑ | Acc ↑ | Score ↑ | Score ↑ |
|---|---|---|---|---|---|---|---|---|---|---|---|---|
| Qwen2.5-VL 3B | 7.5 | **68.0** | 48.8 | 5.0 | **89.6** | 30.7 | 7.6 | **88.6** | **62.5** | 75.9 | **1607.9** | **625.7** |
| DPO | 6.6 | 62.5 | 47.7 | 3.4 | 87.3 | 30.0 | 8.4 | 78.5 | 55.5 | 75.9 | 1558.5 | 616.4 |
| DA-DPO | **6.5** | 57.6 | **42.3** | **2.6** | 88.3 | **25.0** | **7.3** | 80.6 | 56.4 | **76.0** | 1561.5 | 623.9 |

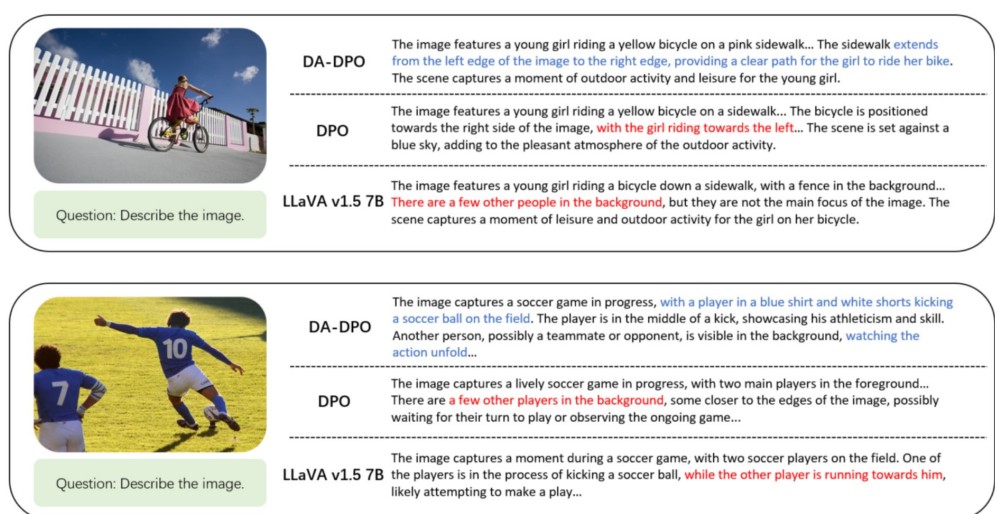

Figure 6: **Visualization of Predictions**. We present the outputs from the proposed method alongside baseline models to highlight the characteristics of our approach. The results are derived from the generative component of AMBER, which is designed to detect hallucinations in image captions.

**Visualization** To provide a better understanding of our method's performance, we present visualizations of the model's outputs, comparing them with the results obtained from both the DPO and reference models in Figure 6. These visualizations highlight the ability of our approach to more effectively reduce hallucinations. By examining the outputs, it is evident that our method aligns better with the expected responses, demonstrating superior accuracy in scene understanding and coherence in generated content.

