# OpenReview forum: "DA-DPO: Cost-efficient Difficulty-aware Preference Optimization for Reducing MLLM Hallucinations"
_TMLR — Accepted by TMLR_

### Review · Reviewer_u6Ks · 2025-09-28

**Summary Of Contributions:**

The paper introduces DA-DPO, a modification to Direct Preference Optimization (DPO) for multimodal large language models (MLLMs). The central idea is to compute a per-pair difficulty score without training any auxiliary models, using two complementary signals: (1) a contrastive image–text gap from a pretrained CLIP model and (2) a generative likelihood gap from a reference MLLM. These normalized signals are fused into a single scalar that scales the DPO logit per example, thereby down-weighting “easy” preference pairs and emphasizing “hard” pairs. The paper argues that this directly addresses an observed tendency of standard DPO to overfit easy pairs, which contributes to visual hallucinations. Experiments across several LLaVA-family models report reduced hallucination rates and smaller capability regressions relative to vanilla DPO. Additional ablations indicate benefits over simple data filtering and show that the two difficulty signals are complementary.

**Additional Comments:**

I think the paper is valuable and with the addition of a few clarifications, would be a good addition to the community. Here are a few additional minor comments -

* **[Minor] Figure 1a readability:** The paper should enlarge Figure 1a and increase the font sizes for its axes and legend. The current graph is difficult to read.
* **[Minor] Define fusion terms on first use:** The weights used in the fusion rule (e.g., `cls_c`, `cls_m`) should be clearly defined upon first use, and the notation should remain consistent throughout the paper.
* **[Minor] Acronyms and capitalization:** All acronyms (e.g., AMBER, POPE, MME) should be defined on first use. Capitalization should be consistent for model names and terms (CLIP, LLaVA, OneVision, VQA, OCR).
* **[Minor] Hyphenation/style pass:** Hyphenation should be standardized (e.g., “cost-efficient,” “per-example,” “state-of-the-art”). The paper would also benefit from a light copy-editing pass for consistency.

**Audience:**

Yes

**Audience Explanation:**

This work addresses a practical, high-impact pain point, multimodal hallucination, via a minimal change to a widely adopted alignment method (DPO). Its training-free difficulty estimation and per-example scaling are straightforward to integrate into existing pipelines, offering a scalable alternative to data filtering that preserves data diversity. The analysis of easy-pair overfitting and the curriculum-like remedy provide conceptual clarity that can inform future work on per-example temperatures, multi-signal difficulty estimation, and robust preference optimization in multimodal systems.

**Claims And Evidence:**

Yes

**Claims Explanation:**

Overall, the paper’s claims are substantiated for the most part, with several focused gaps that could be addressed.

**(C1) Empirically demonstrate the existence of an overfitting issue**

* **What is substantiated:** The paper’s Section 3 presents a training-dynamics analysis showing that models learn more from “easy” pairs than from “hard” ones, using reward trends and relative reward-gap plots as seen in Figure 2 and Figure 6.

* **Where it falls short:**
    * The analysis defines difficulty using only the tails of the CLIP score distribution and discards mid-range cases, which could exaggerate the observed gap-shrinking effect.
    * The paper does not present a classical train-vs-held-out divergence consistent with standard definitions of overfitting.
    * The reward curves appear to be from a single seed without uncertainty bands, so their robustness is unclear.
    * The paper asserts, but does not demonstrate, a within-run link between smaller reward gaps and fewer hallucinations.
    * The evidence for the initial analysis relies on a single proxy (CLIP), which may reflect a bias of that particular scorer.
* **What is needed:** This claim would be better substantiated by analyzing the full difficulty spectrum (e.g., ranked buckets) and varying the cutoff thresholds. The analysis could be strengthened by computing buckets on a held-out split to show easy-vs-hard performance there. For robustness, the paper should report results from $\geq 3$ seeds with mean $\pm$ standard deviation and perhaps a concise area-under-gap summary. To establish a clearer mechanism, the analysis could correlate checkpoint-wise gap reductions to changes in hallucination metrics. Finally, the dynamics analysis should be replicated with an MLLM-based and a fused proxy (ideally with length-controlled likelihoods) to ensure the finding is not proxy-dependent.

**(C2) Propose a cost-effective framework that uses pretrained VLMs to estimate difficulty and improves preference modeling via difficulty-aware training**

* **What is substantiated:** The paper introduces a drop-in DPO modification (Eq. 12) that scales the logit per example using normalized CLIP and MLLM gap scores, fused by a distribution-aware rule. No auxiliary models are trained, and ablations show CLIP-only, MLLM-only, and fused variants, with fusion generally being the strongest approach.

* **Where it falls short:**
    * The “cost-effective” claim is qualitative; the paper provides training times but lacks a full FLOPs or wall-clock accounting for the data scoring and training stages.
    * Fusion weights are learned on the training distribution and may simply mirror its task mix (e.g., caption vs. VQA), rather than capturing true, generalizable reliability. If the weights encode dataset proportions, the “difficulty-aware” gains could fade or flip under a different data mix, weakening the claim that the framework is broadly effective.
    * The effects of normalization choices and text length/verbosity on the difficulty scores are not stress-tested. If the difficulty signals are sensitive to scaling choices or reward longer answers, the method may be optimizing for artifacts of the scoring procedure rather than genuine hardness.
* **What is needed:** This claim needs stronger support. The paper should add a compute table (FLOPs and wall-clock time) for CLIP scoring, MLLM likelihood scoring, and fine-tuning, with a comparison to a baseline like training a small reward model. The fusion mechanism could be validated by re-estimating weights on a held-out split or per-task bucket, alongside ablations with fixed (50:50) weights. The paper should also include ablations on the normalization scheme (e.g., rank/percentile-based) and use length-controlled (per-token) likelihoods for the MLLM score to ensure robustness.

**(C3) Significantly enhance hallucination metrics while preserving (or improving) general ability across multiple MLLMs**

* **What is substantiated:** The paper’s main tables show consistent reductions on hallucination metrics alongside smaller drops (and occasional gains) on comprehensive benchmarks across LLaVA-v1.5-7B/13B and OneVision-7B. Comparisons to filtering baselines favor the proposed reweighting approach over data deletion.

* **Where it falls short:**
    * Statistical and tuning robustness are unclear. The paper appears to report single-seed results. While an appendix table shows a sweep for the DPO temperature ($\beta$), the main comparisons may not be against the optimal baseline setting. Without uncertainty estimates and fairer baseline comparisons, the reported improvements could be due to noise or suboptimal baselines.
    * Generalization and evaluator robustness are not demonstrated. The evidence is confined to the LLaVA model family and a single evaluation setup; alternative model architectures and automatic graders are not examined. The claimed gains may not transfer beyond the tested models or could be dependent on a particular evaluator's artifacts.
* **What is needed:** The claim would be better supported by reporting multi-seed results with uncertainty and paired statistical tests. The paper should include $\beta$ sweeps for vanilla DPO for each model/dataset and ensure comparisons are made against each baseline’s best performance, ideally visualized with metric-versus-$\beta$ plots. To show broader applicability, a small-scale run on a non-LLaVA model (e.g., Qwen-VL/InternVL) would be valuable. Finally, rescoring a subset of outputs with a different automatic evaluator (or a small human audit) would confirm that the improvements are robust to the choice of evaluator.

**Requested Changes:**

1. **[Definite] Statistical robustness:** The paper would be strengthened by providing results over at least three random seeds with uncertainty (mean $\pm$ s.t.d. or confidence intervals) and using paired tests for key metrics.
2.  **[Definite] Baseline tuning fairness:** For transparency and fairness, the paper should include $\beta$ sweeps for vanilla DPO per model and dataset, and compare DA-DPO against each baseline at its best setting. Adding metric-versus-$\beta$ plots would be beneficial.
3.  **[Definite] Cost and efficiency accounting:** To better substantiate the “cost-effective” claim, the paper should include a compute table with FLOPs and wall-clock times for the data scoring and fine-tuning stages, ideally with a comparison to training a small reward model.
4.  **[Definite] Difficulty definition and robustness:** The analysis of difficulty would be more robust if it examined the full difficulty spectrum (e.g., ranked buckets), varied tail cutoffs, constructed buckets on a held-out split, and replicated the dynamics analysis with MLLM-based and fused proxies (using length-controlled per-token likelihoods).
5.  **[Definite] Portability beyond LLaVA:** To demonstrate the method's generality, adding a small-scale experiment on a non-LLaVA model (e.g., Qwen-VL or InternVL) would be a valuable addition.
6.  **[Optional] Mechanistic validation:** To provide deeper insight into the method's mechanics, the authors could log per-example gradient norms to correlate (a) difficulty with gradient magnitude and (b) checkpoint-wise easy–hard gap reductions with changes in hallucination metrics.
7.  **[Optional] Evaluator robustness:** To confirm that the improvements are not an artifact of the evaluator, the paper could include a re-scoring of a subset of outputs with an alternative automatic evaluator or a small-scale human audit.
8.  **[Minor] Presentation and clarity:** For readability, Figure 1a should be enlarged (including axes/legend fonts). The paper should also ensure consistent notation for key terms like $\beta$ and $\hat{\beta}$, define fusion weights on first use, standardize acronyms, and label all axes and units clearly in plots.

---

> ### Author Response · Authors · 2025-10-17
> **Reply for reviewers**
>
> # Reply for the Reviewers' Concerns.
> > ## 1. *Empirically demonstrate the existence of an overfitting issue*
>
> We appreciate the reviewer’s insightful comments. Following the suggestions, we have substantially revised our analysis experiment:
> - **Analysis Experiment Update**: We redesigned the analysis to use a held-out validation set and adopted three distinct difficulty proxies — CLIP-based, MLLM-based, and fused estimations — to ensure robustness against proxy bias. Each experiment is now conducted with three random seeds, and we report both mean and standard deviation to quantify randomness-induced variance. The revised analysis covers the entire difficulty spectrum by partitioning samples into ranked buckets rather than only the tails of the distribution. Detailed settings and results are now presented in Section 3, “Multimodal Preference Optimization Analysis,” of the revised paper.
> - **Correlation Between Reward Gap and Hallucination Reduction**: To directly validate the connection between reward-gap dynamics and hallucination behavior, we designed a new controlled experiment. Specifically, using both CLIP and MLLM proxies, we partitioned the training data into multiple buckets based on the estimated reward gap. Each bucket was then used to train a vanilla DPO model independently. The results consistently show that samples with smaller estimated reward gaps lead to the largest reduction in hallucinations, while the reduction effect diminishes as the gap increases. Detailed results and discussions are provided in Section 5.5, “Estimated Reward Gap and Model Hallucination,” of the revised paper.
>
> >  ## 2. *Propose a cost-effective framework that uses pretrained VLMs to estimate difficulty and improves preference modeling via difficulty-aware training*
>
> Thanks to the reviewer for the suggestions. We have substantially revised our analysis experiment:
> - **Evidence for Cost-effectiveness**: To quantitatively demonstrate the efficiency of our proposed framework, we compared the FLOPs and wall-clock time between the traditional approach of training a full reward model and our estimation-based method. The results show that our method requires only approximately 1/7 of the wall-clock time of the conventional approach. Detailed comparisons can be found in Appendix, Section *Efficiency Comparison*.
> - **Fusion Mechanism**: To validate the effectiveness of the proposed fusion mechanism, we report the per-task classification accuracy of the fused scores in Appendix, Section*Fusion Score Classification Accuracy*. Furthermore, in Appendix, Section*Influence of Adaptive Score Fusion*, we provide an ablation study on different fusion weights. We hope these results provide sufficient empirical support for the design and effectiveness of our fusion mechanism.
> - **Normalization Scheme**: To evaluate the impact of different normalization strategies within our framework, we implemented two alternative approaches—*Ranked-based* and *Length-controlled*—and compared them with the Gaussian normalization used in our main experiments. The results show that all three normalization strategies achieve substantial improvements over the vanilla DPO baseline. We further observe that the Ranked-based approach slightly underperforms the other two strategies, suggesting that preserving the internal distribution of different estimation proxies benefits effective difficulty estimation. The details are provided in Appendix, Section *Difficulty Estimation Normalization Ablation*.

---

> ### Author Response · Authors · 2025-10-17
> **Reply for reviewers**
>
> >  ## 3. *Significantly enhance hallucination metrics while preserving (or improving) general ability across multiple MLLMs*
>
> - **Statistical and tuning robustness**: We appreciate the reviewer’s concern regarding statistical robustness. In the revised paper, Section 3 *Multimodal Preference Optimization Analysis*, we conducted experiments with multiple random seeds and observed that the variations across seeds were minimal, indicating stable optimization behavior. Given the substantial computational cost associated with multimodal large model training, we humbly argue that it is unfortunately infeasible to perform full multi-seed runs for all experimental settings.
> - **Fair Comparison**:
> We appreciate the reviewer’s suggestion regarding parameter tuning. In our initial submission, we adopted the same hyperparameter settings as prior work [1] without extensive tuning. To ensure a fair comparison, we have now conducted a more comprehensive sensitivity analysis of the $\beta$ parameter, as shown in the updated Appendix Section *Sensitivity of $\beta$*. The results show that across more benchmarks, different $\beta$ values lead to mixed results on vanilla DPO. Specifically, $\beta = 0.1$ performs worse only on the AMBER Generative and GQA benchmarks, while remaining competitive on the others. Considering the overall trade-offs and alignment with prior work, we believe that following the previously adopted $\beta = 0.1$ remains a reasonable and fair choice.
>
> - **Different Base Model Choice**:
> To validate the applicability of the DA-DPO framework beyond the LLaVA series, we conducted a small-scale experiment on Qwen2.5-VL. The results show that our method can reduce hallucination metrics while retaining performance on comprehensive benchmarks on this alternative base model. Detailed experimental setup and results are provided in the revised paper Appendix Section  *Experiments with More Model Variants*.
>
>
> # Reply for Request Changes
> 1. We conducted experiments with multiple random seeds in our analysis study, as detailed in the above reply **Analysis Experiment Update**, and observed that the training process exhibits strong robustness across seeds. This is a common phenomenon in post-training settings initialized from well-trained MLLMs. Given the substantial computational cost of multi-seed multimodal training, we believe the current experimental setup provides a reasonable and representative evaluation.
> 2. Please kindly refer to the above reply **Fair Comparison**.
> 3. Please kindly refer to the above reply **Evidence for Cost-effectiveness**.
> 4. Please kindly refer to the above reply **Analysis Experiment Update**.
> 5. Please kindly refer to the above reply **Different Base Model Choice**.
>
> **Reference:**
>
> [1] Strengthening Multimodal Large Language Model with Bootstrapped Preference Optimization.

---

> > ### Comment · Reviewer_u6Ks · 2025-10-21
> > **Response o the rebuttal**
> >
> > I want to thank the authors for addressing the comments that I have raised. With these additional clarifications, I believe the claims made in the paper are substantiated, and the paper is suitable for the TMLR audience, and I will support the paper for acceptance.

---

### Review · Reviewer_T6ss · 2025-10-02

**Summary Of Contributions:**

This paper proposes DA-DPO, a framework for reducing hallucinations in multimodal large language models (MLLMs) by addressing overfitting in Direct Preference Optimization (DPO). The key idea is that MLLMs tend to overfit on “easy” preference pairs, while failing to learn from “harder” and more nuanced examples. To mitigate this, DA-DPO introduces:
* Difficulty Estimation: a training-free method that leverages both contrastive (e.g., CLIP) and generative (e.g., LLaVA) vision-language models to assign difficulty scores to preference pairs, fused via distribution-aware voting.
* Difficulty-Aware Training: reweighting preference data based on difficulty, down-weighting easy samples and emphasizing harder ones, thereby improving robustness without requiring new data or extra fine-tuning stages.

Key strengths:
* Clear identification of a real weakness in current multimodal preference optimization (bias toward easy samples).
* Cost-efficient framework that does not require additional human annotations or training of new models.
* Extensive experiments on multiple hallucination and general benchmarks showing improved robustness and reduced hallucinations.
* Ablation studies confirm the contribution of each component (CLIP vs. MLLM estimators, fusion strategy, filtering baselines).

Key weaknesses:
* Some figures are confusing or hard to read (e.g., Figure 1 and Figure 3).
* Lack of clarity about the proportion of easy/hard/ambiguous samples in the datasets.
* Limited discussion on why DA-DPO’s results are not directly comparable to prior approaches like HA-DPO, CLIP-DPO, or mDPO, although these often show stronger raw results in Table 1.
* Hardware details are incomplete, making reproducibility harder.
* Presentation issues (minor formatting problems, e.g., “LoRA” written with a lowercase o).

**Audience:**

Yes

**Audience Explanation:**

The problem of hallucinations in MLLMs is central to ongoing research, with both theoretical and practical implications. Preference optimization is a rapidly developing area, and a cost-efficient approach that improves robustness without additional supervision is of clear interest to researchers working on alignment, multimodal reasoning, and scalable training methods.

**Broader Impact Concerns:**

The method primarily addresses hallucinations in multimodal models, which is a positive contribution toward more reliable AI. Broader impacts are relatively limited compared to other alignment methods (e.g., RLHF with human data), since the approach relies on existing pretrained models. However, reliance on pretrained VLMs (CLIP, LLaVA) for difficulty estimation may inherit their biases, which could indirectly affect model alignment quality. This limitation should be more explicitly acknowledged.

**Claims And Evidence:**

Yes

**Claims Explanation:**

The paper provides empirical evidence of overfitting on easy samples through reward dynamics analyses. Benchmarks across hallucination (AMBER, MMHal, ObjectHal, POPE) and general MLLM evaluations (SeedBench, GQA, LLaVA-Bench, MME) consistently show that DA-DPO outperforms standard DPO, particularly on hallucination metrics. Ablations (e.g., CLIP-only, MLLM-only, fusion strategies, filtering baselines) convincingly demonstrate that both difficulty estimation and difficulty-aware weighting are necessary. However, the comparability of results with alternative optimization methods (HA-DPO, mDPO) is insufficiently addressed.

**Requested Changes:**

Figures:
* Improve Figure 1 readability (larger font, avoid overlaps).
* Revise Figure 3 to more clearly separate CLIP-based vs. MLLM-based scoring examples.

Clarity:
* Explicitly report how many samples are classified as easy, hard, and ambiguous.

Comparisons and baselines:
* Clarify why HA-DPO, CLIP-DPO, and mDPO results are “not directly comparable,” especially since they achieve higher scores in some benchmarks.

Reproducibility:
* Provide details on hardware (GPUs, memory, batch size) for reported training times.
* Discuss training cost more explicitly to strengthen the claim of “cost-efficient.”

Minor corrections:
* Fix capitalization of “LoRA.”
* Improve caption descriptions in figures to guide interpretation.

---

> ### Author Response · Authors · 2025-10-17
> **Reply for reviewers**
>
> # Reply for Request Changes
>
> 1. **Figures**
> Thanks for the suggestions. We have updated Figure 1 and Figure 3 for clarity.
> 2. **Clarity**
> We have provided a detailed setting section **Analysis Setting**, in the revised paper Section **Multimodal Preference Optimization Analysis**.
> 3. Comparisons and baseline.
> We would like to clarify that existing multimodal DPO works (e.g., HA-DPO, CLIP-DPO, mDPO) are built upon different data settings and training configurations. Our experiments primarily follow the BPO [1] setup to ensure a controlled evaluation of our framework’s effectiveness. Moreover, our method is orthogonal to approaches such as mDPO and could potentially be combined with them in future work. Given the limited computational budget, our current focus is to validate the effectiveness of the proposed framework itself rather than exhaustively reimplementing alternative settings.
> 4. **Reproducibility**
> We have provided an efficiency comparison in Appendix **Efficiency Comparison** to further support the claim of cost efficiency. The full training details will be released with the code upon acceptance.
> 5. **Minor Corrections**
> Thanks for the suggestion. We have revised the paper accordingly.
>
>
>
> **Reference:**
>
> [1] Strengthening Multimodal Large Language Model with Bootstrapped Preference Optimization.

---

### Review · Reviewer_fFru · 2025-10-03

**Summary Of Contributions:**

The paper presents DA-DPO, which leverages external difficulty proxies to dynamically steer the divergence penalty strength in the DPO algorithm. With CLIP and multi-modal large language models (MLLMs) as the two difficulty proxies, the authors propose the score margin for the chosen and rejected responses given the image as a heuristic to determine the appropriate divergence penalty strength. The experimental results are shown across different sizes and models on multiple benchmarks, both for hallucinations and comprehensive benchmarks. While the paper provides comprehensive experiments, the motivation and method remain unclear. Overall, the paper is promising in direction and provides valuable empirical results, but would benefit from clearer motivation and a more rigorous treatment of the difficulty concept.

**Audience:**

Yes

**Audience Explanation:**

The paper's approach clearly has empirical value, especially regarding its wide scope of evaluation from hallucination-specific benchmarks to comprehensive ones. Also, the direct alignment algorithms that are specifically targeted to MLLMs are not yet widely studied compared to the plain language models. Thus, the empirical results can draw the interest of the audience.

**Broader Impact Concerns:**

No specific concerns regarding the ethical implications.

**Claims And Evidence:**

No

**Claims Explanation:**

1. **Unclear connection between "*difficulty score*" and preference difficulty**: As the title suggests, DA-DPO builds on an assessment of sample “difficulty” in multimodal preference learning. The paper introduces two proxies: (1) the CLIP-score margin (cosine similarity margin between chosen and rejected responses given the image), and (2) the MLLM log-likelihood margin. However, their interpretation remains unclear.

   - Why should a larger CLIP-score margin imply easier distinguishability, and in turn, why does this make the sample an easier instance for preference optimization?
   - In what sense does the MLLM log-likelihood margin reflect difficulty?

   While the high-level intuition is understandable, the connection to preference learning is neither theoretically grounded nor empirically substantiated. For example, the CLIP margin reflects image–text relevance, which does not necessarily capture the implicit reward separability discussed in Section 3. Similarly, the log-likelihood margin $m_c - m_r$ may serve as a proxy under reference-free algorithms such as SimPO [1] or ORPO [2], where it aligns with the logistic loss term. But in DPO, the implicit reward $\log \frac{\pi\_\theta(y|x)}{\pi\_\mathrm{ref}(y|x)}$ always starts with zero, making the connection less direct. A stronger theoretical or empirical justification is needed, and the manuscript currently does not cite related work that could support this interpretation.

2. **Missing analysis on the reward dynamics and over-optimization**: Section 3 motivates DA-DPO by suggesting that standard DPO can suffer from over-optimization (or overfitting) when difficulty variance is high, and that difficulty-aware weighting mitigates this. However, the analysis presented is based on training-set reward trajectories. To substantiate the claim, evaluation on held-out or validation sets would be essential [3-5]. Without such evidence, it remains unclear whether DA-DPO actually alleviates over-optimization in the sense of improved generalization, rather than simply altering training dynamics.

3. **Additional questions**
   - In Table 1, results across the “Main Results” section appear mixed, especially on comprehensive benchmarks. Could the authors provide more insight into why these tendencies arise?
   - In Appendix C, the authors report DA-DPO performances with varying $\beta$. However, as in Equation (11), $\hat{\beta}$ is a dynamic function of difficulty scores and the static $\beta$ is not directly present in $\mathcal{L}_\mathrm{DA-DPO}$. How should this ablation be interpreted?
   - Why is the method specifically targeted at hallucination mitigation? The method itself is rather designed for addressing the difficulty-awareness in DPO, not hallucination-specific, especially in Sections 3 to 4. Is it because the gains are more notable in the hallucination benchmarks?

&nbsp;

**References**

[1] Meng et al., 2024, "SimPO: Simple Preference Optimization with a Reference-Free Reward."

[2] Hong et al., 2024, "ORPO: Monolithic Preference Optimization without Reference Model."

[3] Gao et al., 2023, "Scaling Laws for Reward Model Overoptimization."

[4] Rafailov et al., 2024, "Scaling Laws for Reward Model Overoptimization in Direct Alignment Algorithms."

[5] Shi et al., 2024, "Understanding Likelihood Over-optimisation in Direct Alignment Algorithms."

**Requested Changes:**

1. **Clarify the definition of difficulty**: Provide theoretical or empirical justification for why CLIP-score and MLLM log-likelihood margins correspond to preference difficulty in DPO.
2. **Over-optimization analysis**: Add validation or held-out results to demonstrate that DA-DPO mitigates over-optimization in generalization, not just in training dynamics.
3. **Interpretation of $\beta$ ablation**: Clearly explain how varying $\beta$ in Appendix C interacts with Equation (11), and what the reported ablation actually measures.
4. **Empirical support for difficulty proxies**: Include correlation or binning studies showing how the proposed difficulty scores relate to optimization success or generalization error.
5. **Mixed benchmark results**: Offer analysis or discussion of why DA-DPO improves some benchmarks but not others.
6. **Clearer connection between the method and hallucination**: While the title emphasizes hallucination mitigation, it is hard to find the connection between hallucination and the core motivation/method of the paper; it can only be found in the experiments.

---

> ### Author Response · Authors · 2025-10-17
> **Reply for reviewers**
>
> # Reply for the Reviewers' Concerns.
> > ## 1. **Unclear connection between "difficulty score" and preference difficulty**
>
> We thank the reviewer for raising this point and appreciate the opportunity to clarify our motivation.
>
> **(1) Definition of preference difficulty.**
> We first clarify that, for pairwise preference data, **preference difficulty** is defined as the gap between the chosen and rejected examples: pairs whose chosen and rejected responses are very similar exhibit a *small gap* and are typically *high-quality* (i.e., low difficulty), whereas pairs with a larger gap indicate greater separability and are correspondingly easier for preference learning.
>
> **(2) Rationale for the proposed proxies.**
> Our two proxies—**CLIP-score margin** and **MLLM log-likelihood margin**—approximate this gap from complementary perspectives:
> - The **CLIP-score margin** measures the relative image–text alignment between the two responses, offering a *contrastive* estimation of preference separability grounded in pre-trained multimodal understanding.
> - The **MLLM log-likelihood margin** captures the model’s *generative* confidence difference between the two responses, reflecting the model’s intrinsic view of preference distinguishability.
> Both indicators leverage existing model priors without introducing additional computational cost, providing lightweight yet informative approximations of the latent gap between chosen and rejected response in pairwise preference data.
>
> **(3) Empirical verification.**
> To strengthen the empirical connection, we have added a new experiment in the revised paper (Section 5.5, *“Estimated Reward Gap and Model Hallucination”*). Specifically, we partition the training data into multiple buckets according to the estimated reward gap using both CLIP and MLLM proxies. For each bucket, we train a vanilla DPO model independently. The results show a positive correlation between the estimated difficulty score and actual training behavior: data with lower difficulty scores (i.e., larger reward gaps) lead to lower model performance.
>
> These findings provide strong empirical support that our proposed difficulty estimators meaningfully capture preference difficulty in multimodal preference learning.
>
> > ##  2. **Missing analysis on the reward dynamics and over-optimization**
>
> We redesigned the analysis to use a held-out validation set and adopted three distinct difficulty proxies — CLIP-based, MLLM-based, and fused estimations — to ensure robustness against proxy bias. Each experiment is now conducted with three random seeds, and we report both mean and standard deviation to quantify randomness-induced variance. The revised analysis covers the entire difficulty spectrum by partitioning samples into ranked buckets rather than only the tails of the distribution. Detailed settings and results are now presented in Section 3, “Multimodal Preference Optimization Analysis,” of the revised paper.

---

> ### Author Response · Authors · 2025-10-17
> **Reply for reviewers**
>
> > ##   **3. Additional questions**
> 3.1 **Mixed results.**
> We appreciate the reviewer’s observation. The mixed results on comprehensive benchmarks mainly **reflect the trade-off between hallucination mitigation and general ability retention** in our small-scale post-training setup.
>
> Because **the preference data is tailored to hallucination reduction**, the model is naturally encouraged to prioritize factual consistency and grounding over broader task generalization. In such a limited post-training regime, this targeted optimization may also induce **partial forgetting of previously acquired general knowledge**, leading to the observed mixed tendencies. Nevertheless, as shown in Table 1, DA-DPO achieves a favorable overall balance, substantially mitigating hallucinations while retaining competitive general performance.
>
> This phenomenon is closely related to the **knowledge forgetting issue in continual learning**, which we leave to future work for further exploration toward stabilizing knowledge retention while preserving DA-DPO’s strong hallucination robustness.
>
> 3.2 **Interpretation of Objective Function.**
> Sorry for the confusion. The \(\mathcal{L}_{\text{DA-DPO}}\) in Equation (12) indeed involves the parameter \(\beta\), which is embedded in the definition of \(r(x, y)\) in Equation (2). In the revised version, we have made this dependence explicit by rewriting Equation (12) to clearly show how \(\beta\) contributes to the loss.
>
>
> 3.3 **Why Target at Hallucination.**
> Our method is **not specifically designed for hallucination mitigation**, but rather serves as a **general difficulty-aware DPO framework** that can be applied to various preference learning tasks. We focus on hallucination benchmarks mainly due to **data and proxy availability** instead of any methodological limitation.
>
> First, the **available preference data** are primarily constructed for hallucination and faithfulness supervision, which provides a foundation to validate the proposed framework. Second, one of our **difficulty estimation proxies**, the CLIP-based image–text relevance score, is naturally correlated with hallucination difficulty—samples with lower alignment tend to be more hallucination-prone, making this proxy particularly suitable for our study.
>
> In principle, if **task-specific preference data** and **appropriate difficulty estimators** become available for other domains, the same framework can be directly extended and evaluated in those settings. We plan to explore such broader applications in future work.
>
> # Reply for Request Changes
> 1. Please kindly refer to the above reply **Unclear connection between "difficulty score" and preference difficulty**.
> 2. Please kindly refer to the above reply **Missing analysis on the reward dynamics and over-optimization**.
> 3. Please kindly refer to the above reply **Interpretation of Objective Function.**
> 4. Please kindly refer to the above reply **Unclear connection between "difficulty score" and preference difficulty**.
> 5. Please kindly refer to the above reply **Mixed results.**
> 6. Please kindly refer to the above reply **Why Target at Hallucination.**

---

### Decision · Action_Editor_bst1 · 2025-11-26

**Recommendation:** Accept as is

**Audience:**

Yes

**Audience Explanation:**

This paper should be of interest to people working on topics related to hallucinations of LLMs, preference optimization, and data selection / reweighing for machine learning.

**Claims And Evidence:**

Yes

**Claims Explanation:**

The overall empirical evidence on benchmark datasets is strong and coherent. After the rebuttal, the submission added new experiments on the connections between the difficulty score and preference difficulty, used held-out sets of showcase overfitting dynamics, added results on a different base model, and report wall-clock time. The empirical claims are consistent across multiple benchmarks, with proper ablations.